# DENOISING DIFFUSION STEP-AWARE MODELS

**Shuai Yang** [1,3]   **Yukang Chen** [2]   **Luozhou Wang** [1,3]

**Shu Liu** [4*]   **Yingcong Chen** [1,5*]

[1]HKUST(GZ)   [2]CUHK   [3]HKUST(GZ) - SmartMore Joint Lab
[4]SmartMore   [5]HKUST

## ABSTRACT

Denoising Diffusion Probabilistic Models (DDPMs) have garnered popularity for data generation across various domains. However, a significant bottleneck is the necessity for whole-network computation during every step of the generative process, leading to high computational overheads. This paper presents a novel framework, Denoising Diffusion Step-aware Models (DDSM), to address this challenge. Unlike conventional approaches, DDSM employs a spectrum of neural networks whose sizes are adapted according to the importance of each generative step, as determined through evolutionary search. This step-wise network variation effectively circumvents redundant computational efforts, particularly in less critical steps, thereby enhancing the efficiency of the diffusion model. Furthermore, the step-aware design can be seamlessly integrated with other efficiency-geared diffusion models such as DDIMs and latent diffusion, thus broadening the scope of computational savings. Empirical evaluations demonstrate that DDSM achieves computational savings of 49% for CIFAR-10, 61% for CelebA-HQ, 59% for LSUN-bedroom, 71% for AFHQ, and 76% for ImageNet, all without compromising the generation quality. Our code and models are available at `https://github.com/EnVision-Research/DDSM`.

## 1 INTRODUCTION

Denoising diffusion probabilistic models (DDPMs) have emerged as a powerful tool in the generation of high-quality samples across a range of media. The core idea hinges on a predefined diffusion process that gradually corrupts the original information by injecting Gaussian noise into the data. Subsequently, a neural network learns a denoising procedure to systematically recover the original data step by step.

However, a notable drawback of DDPMs lies in their efficiency. The generative phase requires hundreds to thousands of iterations to reverse the noise and retrieve samples, significantly lagging behind the efficiency of other generative models like Generative Adversarial Networks (GANs) that only necessitate a single pass through the network. Moreover, the iterative nature of diffusion models requires a full pass through the neural network at every denoising step to yield a single sample, raising questions regarding its efficiency and practicality.

Given these challenges, a natural question arises: do all diffusion steps require equal computational resources? We hypothesize that different diffusion steps have varying levels of importance. In certain less critical steps, a smaller model might suffice. It is, therefore, imprudent to allocate equal computational resources across all steps. To substantiate this hypothesis, we crafted a fusion of a

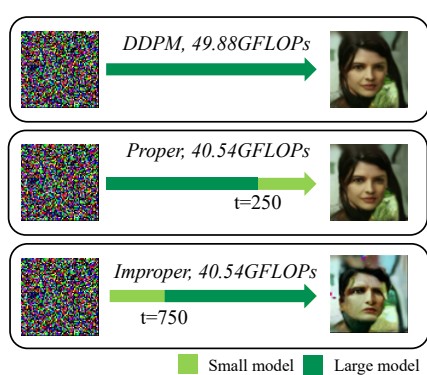

Figure 1: Combinations of different models along steps. The proper combination leads to high-quality samples and computational saving.

---

*Corresponding Author

large and a small model into four distinct model combinations for various steps. Our findings reveal a notable variance in performance across different combinations, even under identical computational budgets. As illustrated in Figure 1, a good model combination produces favorable outcomes, whereas an inappropriate one leads to a less pleasing generation. This insight underscores the fact that the significance varies across different steps, pinpointing redundancy in contemporary DDPMs. Thus, the quest for an astute step-aware model combination can help pare down redundancies and expedite the diffusion model's operations.

Based on these observations, we propose using different networks for different steps. We delve into the formulation of diffusion models and extend it to a more general one. We show the feasibility of this approach and present the Denoising Diffusion Step-aware Models (DDSM). In DDSM, the neural network is variable and slimmable at different steps to avoid redundant computation at unimportant steps. We determine the capacity of the neural network at each step via evolutionary search and prune the network to various scales accordingly. The pruned network requires no re-training and shares weights with the intact one.

In our comprehensive empirical evaluation, we showcase the benefits of DDSM by conducting extensive experiments on five distinct datasets: CIFAR-10, CelebA-HQ, LSUN-bedroom, AFHQ, and ImageNet. We successfully achieved FLOPs acceleration rates of **49%** for CIFAR-10, **61%** for CelebA-HQ, **59%** for LSUN-bedroom, **71%** for AFHQ, and **76%** for ImageNet. The search result varies according to the attributes of different datasets, further indicating the significance of step-aware search.

In addition, our findings indicate that DDSM, a unique method focusing on pruning network steps, is orthogonal and easily compatible with established diffusion acceleration techniques like DDIM (Song et al., 2020a) and latent diffusion (Rombach et al., 2022). Our experiments show that integrating DDSM can further boost their efficiency."

## 2 RELATED WORK

**Diffusion Models**    Diffusion models (Sohl-Dickstein et al., 2015; Song & Ermon, 2019; Song et al., 2020b; Ho et al., 2020; Nichol & Dhariwal, 2021; Vahdat et al., 2021) are generative models that transform a simple Gaussian distribution into a more complex data distribution. This is done progressively, enabling the generation of high-quality samples from the target distribution. However, a major concern with the diffusion model is the high computational cost resulting from the repeated use of a large U-net model, which typically consists of more than 35.75M parameters and requires 12.14T FLOPs to produce a sample for CIFAR-10.

**Diffusion Acceleration**    Several existing works have sought to mitigate the computational costs associated with the extended step. For instance, DDIM (Song et al., 2020a), PNDM (Liu et al., 2022), and DPM solver (Lu et al., 2022a;b) have introduced first-order and high-order solvers. These solvers drastically cut down the number of steps required during inference. Additionally, knowledge distillation techniques have been employed by researchers (Salimans & Ho, 2022; Luhman & Luhman, 2021) to train a student model from a teacher model, effectively reducing the step. Another innovative approach involves compressing the data prior to training the denoising model (Rombach et al., 2022). By working with a reduced dimension of input data, this method diminishes the computational demands of inference. AutoDiffusion (Li et al., 2023), a concurrent study, suggests a non-uniform skipping of steps and blocks within the generation sequence, offering further acceleration to DDPMs. In our study, we tackle the challenge from a unique angle. We aim to curtail the overall computation of the diffusion generation process by adaptively pruning the U-net at each step. It's important to highlight that our strategy is orthogonal to the aforementioned methods and can seamlessly integrate with them. Several notable works share a close relationship with ours. OMS-DPM (Liu et al., 2023) firstly proposes an effective predictor-based algorithm to optimize the model-schedule. Compared to OMS-DPM, our work advances this field by implementing a supernet, efficiently scaling up the model zoo to a considerably larger size. Spectral Diffusion (Yang et al., 2022), endeavors to expedite diffusion models by harnessing denoising loss to train a dynamic gate for acceleration. Our method, however, adopts a more straightforward metric: the FID. While Spectral Diffusion incorporates channel filtering, leading to sparse feature maps, we utilize slimmable networks and channel slicing. This results in continuously pruned channels, optimizing GPU efficiency. While eDiff-I Balaji et al. (2023) adopts multiple expert models at each step to improve

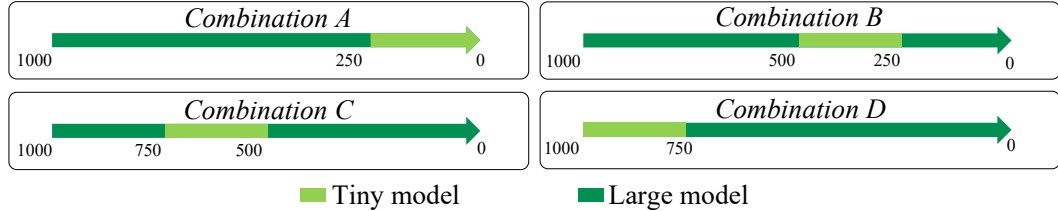

Figure 2: The diagrams of different model combinations. Combinations A, B, C, and D have the same computation cost. The only difference between them is the steps of using tiny models.

image quality, it does not enhance efficiency, as all these expert models are uniformly sized. Our method distinguishes itself by utilizing models of varying sizes for acceleration purposes.

**Network Pruning** Model pruning has a rich history in the literature of neural networks. In the early period, model pruning (Han et al., 2016) is limited to sparsify connections in the network architecture, while the actual speedup from these methods might be constrained by hardware. Latter methods (Ye et al., 2018; Luo et al., 2017; Liu et al., 2017) extend the pruning scope to channels, filters, or layers. Slimmable network (Yu & Huang, 2019b) further includes knowledge distillation into pruning. It trains a supernet network that is executable at various widths, which enables adaptive accuracy-efficiency trade-offs during inference. Network pruning focuses on the architecture in a single inference time while ours makes it to be step-aware in various denoising steps.

## 3 DISCUSSION OF STEP-AWARE NETWORKS

### 3.1 PILOT STUDY

Conventional diffusion models use a heavy network for all denoising steps, ignoring the differences between the steps. However, we hypothesize that some steps in the generation process may be easier to process, and that even a lightweight network can handle these steps. Treating all steps equally leads to redundancies.

To verify our hypothesis, we conducted experiments on CIFAR-10, with two pre-trained networks of different model sizes. We first tried replacing all steps with smaller models. As shown in line 1 and line 2 of Table 1, the large model obviously outperformed the tiny one. Generally, the large model demonstrates its stronger ability to denoising. Uniform pruning to the tiny model leads to drops in image quality.

However, when we only prune some specific steps of the process, the results seem to vary. We manually composed four model combinations with both the large and tiny models and evaluated their performance. To ensure the same computation cost, all combinations used the large model for 750 steps and the tiny model for 250 steps, but the range of steps using tiny models varied across the combinations. Combinations A, B, C, and D employed tiny models for the step ranges [0, 250), [250, 500), [500, 750), and [750, 1000), respectively. The diagrams of these combinations are shown in Figure 2.

Table 1: Evaluation of the model combinations on CIFAR-10.

| Method | FID | GFLOPs |
|---|---|---|
| DDPM-tiny | 18.64 | 0.38 |
| DDPM-large | 3.71 | 12.1 |
| Combination A | 10.61 | 9.20 |
| Combination B | 4.38 | 9.20 |
| Combination C | 3.30 | 9.20 |
| Combination D | 3.62 | 9.20 |

The surprising evaluation results showed that different combinations led to different outcomes, as shown in Table 1. Proper model combinations achieved comparable, or even slightly better, image quality than the baseline large model, while taking less computation cost. Improper model combinations led to significant drops in performance, even though they had the same FLOPs as the proper ones.

These experiments validated our hypothesis and gave us three conclusions:

- It is not necessary to use large models for every step.

- The importance of different diffusion steps varies.
- If we fully utilize the tiny model for its suitable steps, we can achieve excellent results with less computational cost.

These exciting conclusions have led us to a way to improve the diffusion model efficiency. We are eagerly seeking an excellent step-aware strategy to remove redundancies and accelerate the model.

## 3.2 FORMULATION ANALYSIS

In this section, we show that it is possible to combine different networks in a diffusion process. We first review the mathematical model of the diffusion process, and then expand it to a generalized version that supports step-aware diffusion.

Diffusion models are generative models that use latent variables. They define a Markov chain of diffusion steps that gradually add noise to data, and then train a neural network to approximate the reverse diffusion distribution to construct desired data samples from the noise (Sohl-Dickstein et al., 2015; Ho et al., 2020).

Suppose the data distribution is $q(x_0)$ where $x_0$ denotes the original data. Given a training data sample $x_0 \sim q(x_0)$, the forward diffusion process aims to produce a series of noisy latents $x_1, x_2, \cdots, x_T$ by the following Markovian process,

$$q(x_t \mid x_{t-1}) = \mathcal{N}(x_t; \sqrt{1 - \beta_t} x_{t-1}, \beta_t \mathbf{I}), \forall t \in T, \tag{1}$$

where $t$ is the step number of the diffusion process, $T$ denotes the set of the steps, $\beta_t \in (0, 1)$ represents the variance in the diffusion process, $\mathbf{I}$ is the identity matrix with the same dimensions as the input $x_0$, and $\mathcal{N}(x; \mu, \sigma)$ means the normal distribution with mean $\mu$ and covariance $\sigma$.

To generate a new data sample, diffusion models sample $x_T$ from a standard normal distribution and then gradually remove noise using the intractable reverse distribution $q(x_{t-1} \mid x_t)$. Traditionally, diffusion models learn a general neural network $p_\theta$ for all steps to approximate the reverse distribution, as shown in Eq. 2.

$$p_\theta(x_{t-1} \mid x_t) = \mathcal{N}(x_{t-1}; \mu_\theta(x_t, t), \Sigma_\theta(x_t, t)), \tag{2}$$

where $\mu$ and $\Sigma$ are the trainable mean and covariance functions, respectively.

It is natural to expand this to use a different network $\theta$ for different steps. The reverse distribution does not necessarily require the same network for the entire process, as long as the network can approximate the reverse distribution $p$. To release this constraint, we only need to change the network $\theta$ in Eq. 2 to a functional $F(t)$, as shown in Eq. 3. The $F(t)$ returns different networks $\theta$ for different steps $t$. In this formulation, the traditional method can be viewed as a special case of ours, where $F(t)$ is a constant functional that returns the same function $\theta$ for all steps.

$$p_{F(t)}(x_{t-1} \mid x_t) = \mathcal{N}(x_{t-1}; \mu_{F(t)}(x_t, t), \Sigma_{F(t)}(x_t, t)), \tag{3}$$

Therefore, with the expanded version of the diffusion model in Eq. 3, it is theoretically possible to combine different networks in a diffusion generation process.

## 4 STEP-AWARE NETWORK FOR DIFFUSION MODELS

### 4.1 STEP-AWARE NETWORK TRAINING

In Section 3.1, we show that a human-designed model combination of two different sub-networks with various model sizes can improve efficiency. However, this design still has two constraints. First, the design's flexibility is limited by the number of sub-networks. Fewer sub-networks may not provide enough flexibility to adjust network size for different steps, while more sub-networks would require training each of them, significantly increasing training costs. Second, a human-designed ensemble strategy may result in sub-optimal results.

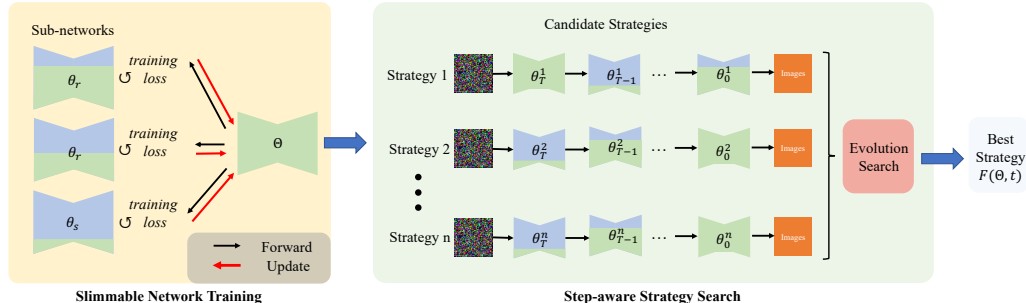

Figure 3: The overall procedure of DDSM, including training and searching. In this figure, the color green represents the activated weights within the network, whereas blue represents the inactive weights. The varying proportions of blue and green in different networks indicate networks of various sizes. The supernet, which has all of its weights activated, is depicted entirely in green. In the training stage, we update the slimmable network via its sub-networks optimization. When the slimmable network converges, it can be executed at any arbitrary size. In the searching stage, we aim to find a step-aware strategy that assigns different models to their suitable steps. We employ the evolution algorithm to search for the optimal strategy.

---

**Algorithm 1** DDSM Training

---

1: $q(x_0)$ is the data distribution, $S$ is the the step set, $\bar{\alpha}_t$ is the noise schedule parameter.
2: $\Theta$ is the slimmable network, $\theta_l$ and $\theta_s$ is the largest and the smallest sub-network of $\Theta$.
3: **repeat**
4:     $x_0 \sim q(x_0)$, $t \in S$, $\epsilon \sim \mathcal{N}(\mathbf{0}, \mathbf{I})$
5:     $\theta_r \sim \Theta$ // Sample a random sub-network
6:     **for** $\theta = \theta_l, \theta_s, \theta_r$ **do**
7:         Take gradient descent step on
8:         $\nabla_\phi \left\| \epsilon - \epsilon_\theta(\sqrt{\bar{\alpha}_t}x_0 + \sqrt{1 - \bar{\alpha}_t}\epsilon, t) \right\|^2$
9:     **end for**
10: **until** $\Theta$ converged

---

To address these constraints, we introduce the slimmable network and the step-aware evolutionary search. A slimmable network (Yu et al., 2018; Yu & Huang, 2019b;a) is a neural network that can be executed at arbitrary model sizes. When training the slimmable network, we simultaneously push up the lower and upper bounds of its sub-networks, which can implicitly boost the performance of all sub-networks.

Specifically, for each training iteration, we sample a random sub-network $\theta_r$ from the slimmable network $\Theta$. Then, we take the DDPM training step on the largest sub-network $\theta_l$, the smallest sub-network $\theta_s$, and the sampled sub-network $\theta_r$. By optimizing these sub-networks, we can get a slimmable network that contains a large number of networks with fine-grained model size options, without the need for multiple training runs. The training procedure for DDSM slimmable network is shown in Algorithm 1.

Our method applies the slimmable network to diffusion denoising tasks instead of image recognition. Through empirical testing, we have found that the in-place distillation and switchable batch normalization, which were proposed by the slimmable network, do not improve network performance on the diffusion task. Therefore, we have removed these techniques for simplicity.

## 4.2 STEP-AWARE EVOLUTIONARY SEARCH

After training the slimmable network, we perform a multi-objective evolutionary search within the slimmable network to minimize the model size and maximize performance under different computational constraints. NSGA-II (Deb et al., 2002) is the most commonly used method for this type of search (Jozefowicz et al., 2015; Pietroń et al., 2021; Elsken et al., 2019; Guo et al., 2020) due

---

**Algorithm 2** DDSM Evolutionary Searching

---

1: **Input: the slimmable network** $\Theta$**, number of generation** $G$**, population size** $P$**, mutation probability** $m$**, FLOPs objective weight** $w_M$**, source dataset** $X$
2: Generate the initial population p of strategies
3: **for** g in range(G) **do**
4:     **for** each individual i in p **do**
5:         score(i) = FID(i, $\Theta$, $X$) + $w_M \times$ FLOPs(i)
6:         assign a rank to the individual i
7:     **end for**
8:     selection(p, P) # select new population
9:     crossover(p) # crossover among individuals
10:    mutation(p, m) # mutate individuals according to m
11: **end for**
12: find the best individual $F$ with the highest score
13: **Output: the best step-aware strategy** $F$

---

to its non-dominated sorting method, which helps balance conflicting objectives. This objective-oriented method eliminates the need for extensive human efforts in step-aware strategy designing and demonstrates a strong ability in finding solutions.

To conduct the search, we use NSGA-II as the search agent. We initialize the NSGA-II with some uniform non-step-aware strategies and some random strategies in the first generation. Our search algorithm adopts single-point crossover on a random position and random choice mutation to enhance generation diversity. When a mutation occurs, a selected step model size changes to another available choice randomly. The searching algorithm can be seen in Algorithm 2.

Once we obtain an optimal step-aware strategy $F$, we can use it to speed up the sampling process. For each sampling, we employ our step-aware strategy $F$ to select the most suitable sub-network $\theta$ from the slimmable network $\Theta$, and we use $\theta$ to denoise a single diffusion step. The overall procedure of DDSM is presented in Figure 3, including training and searching.

### 4.3 COMPATIBILITY WITH OTHER ACCELERATION

Our method accelerates diffusion models by using different-sized models to process different steps. It is compatible with existing methods, making it a plug-and-play module that is easy to use. When combined with latent diffusion models (Rombach et al., 2022), we only need to train the slimmable network on the latent space. When combined with step-skipping methods (Song et al., 2020a; Liu et al., 2022; Lu et al., 2022a;b), we only need to reduce the evolutionary search space to the skipped one.

Our experiments in Section 5.4 show that our method works well with DDIM and Latent Diffusion, producing a much more efficient diffusion model compared to the baselines. This makes our approach effective and practical. It provides a new angle to accelerate diffusion while keeping quality high and computational costs low.

## 5 EXPERIMENT

We conduct experiments on five image datasets on different domains, ranging from small scale to large scale. They are CIFAR10 (Krizhevsky et al., 2009), CelebA-HQ (64x64, 128x128) (Liu et al., 2015), LSUN-bedroom (Yu et al., 2016), AFHQ (Choi et al., 2020), and ImageNet (Deng et al., 2009). Due to the length limitation, we present the LSUN-bedroom, AFHQ, CelebA-HQ-128, and conditional CIFAR-10 results in the appendix.

### 5.1 IMPLEMENTATION DETAIL

**Data settings** For the CIFAR-10 dataset, we utilized the standard split of 50,000 images designated for training. These images were normalized and resized to a resolution of 32x32 pixels. For the CelebA-HQ-64, AFHQ, and LSUN-bedroom datasets, we adhered to the typical training data

splits, normalizing and resizing the images to 64x64 pixels. As for ImageNet, we aligned our processing with the unconditional ImageNet-64 settings as described in Nichol & Dhariwal (2021). During training for all these datasets, we incorporated random horizontal flipping as a data augmentation technique.

**Training settings** Our network architecture followed the setting in ADM (Dhariwal & Nichol, 2021), without the classifier guidance. We used a U-Net (Ronneberger et al., 2015) with an attention mechanism to down-sample input images and then up-sample them back. For more training details, we present them in the Section D of the appendix.

**Search and evaluation** For our search algorithm, we utilized NSGA-II (Deb et al., 2002), and implemented it leveraging the open-source tools pymoo (Blank & Deb, 2020) and pytorch (Paszke et al., 2017). For assessing image quality, we adopted the FID metric (Heusel et al., 2017). To evaluate speed, we calculated the average Floating Point Operations (FLOPs) across all steps throughout the image generation process. Additionally, we assessed real GPU latency to demonstrate the genuine acceleration capabilities of our method. The GPU latency is the time cost of generating one image with a single NVIDIA RTX 3090. For more details about search and evaluation, we present them in the Section D of the appendix.

Table 2: A comparative analysis of ADM and DDSM models on CIFAR-10, CelebA-HQ, and ImageNet datasets. GLOPs are measured as the average over 1,000 generation steps, and latency represents the GPU processing time (in seconds) for generating a single image on an NVIDIA RTX-3090.

| method | CIFAR-10 | | | CelebA-HQ | | | ImageNet | | |
|---|---|---|---|---|---|---|---|---|---|
| | FID | GFLOPs | latency | FID | GFLOPs | latency | FID | GFLOPs | latency |
| ADM | 3.713 | 12.14 | 0.93 | 6.755 | 49.88 | 2.25 | 27.000 | 82.06 | 4.75 |
| **DDSM** | **3.552** | **6.20** | **0.59** | **6.039** | **19.58** | **1.13** | **24.714** | **19.40** | **3.22** |

## 5.2 RESULT

Table 2 shows the quantitative results of our DDSM compared to ADM, on CIFAR-10, CelebA-HQ, and ImageNet. Our DDSM shows its superiority on speed while maintaining high generative quality. It is also noteworthy that our method not only reduce the FLOPs of diffusion inference, but it also achieve actual acceleration. When pruning the networks with the step-aware strategy, we slice the channel contiguously. This is a GPU friendly operation and could provide actual acceleration.

On CIFAR-10 and ImageNet, DDSM has only **51%** and **24%** FLOPs of the ADM but still achieves better image quality. On CelebA-HQ, DDSM only takes about **39%** FLOPs of the original ADM, yet it produces images with comparable quality.

## 5.3 ABLATION STUDY ON THE STEP-AWARE STRATEGIES

Table 3: Comparison of different step strategies. The experiments are conducted on CIFAR-10. The result of the random strategy is calculated by averaging three different random strategies.

| Strategy | FID | avg. GFLOPs |
|---|---|---|
| ADM-large | 3.713 | 12.14 |
| ADM-mid | 4.697 | 8.84 |
| ADM-small | 5.936 | 3.04 |
| ADM-tiny | 18.638 | 0.38 |
| Random | 7.727 | 5.64 |
| **DDSM** | **3.552** | **6.20** |

In this section, we compared DDSM to different strategies. The ADM-large denotes the most common baseline ADM, which takes the whole network to sample an image. The ADM-mid, ADM-small and ADM-tiny uniformly prune the network for each of the steps with different pruning densities. They still take the same network to denoise and treat all steps equally. The random strategy means we completely choose a random model size for each step. DDSM denotes the strategy searched by our proposed method.

The result is shown in Table 3 The ADM-large takes heavy average FLOPs of 12.14 to produce an image. The ADM-mid, ADM-small

and ADM-tiny manage to reduce their computational cost, but they scarify image quality. The random strategy even achieves worse results than the ADM-mid strategies, indicating that an aimless combination of networks without step awareness could even decrease the performance. Compared to all of these baselines, our DDSM strategy demonstrates its superiority in both image quality and speed.

## 5.4 COMBINING WITH OTHER ACCELERATION

In this section, we present our work's compatibility with the step-skipping acceleration methods and latent diffusion. We choose the most commonly used scheduler, DDIM and a recent sampler EDM to conduct experiments. More details can be found in our appendix.

DDIM models the denoising diffusion process in a non-Markovian process and proposes to respace the sampling procedure. It skips the steps of the diffusion generation process while generating high-quality images. Our method is theoretically compatible with DDIM since we only focus on model width rather than steps. To integrate DDIM, we searched the step-aware strategy on the respaced time steps. We employed DDSM in different step-skipping degrees of DDIM, as shown in Table 4. It shows that our method achieves further acceleration compared to separately using DDIM, no matter what step-skipping degree DDIM adopts.

Table 4: Combining DDSM with DDIM. The upper part of the table presents the FID score. The lower part of the table presents the total FLOPs of the generation process. Our DDSM can further accelerate DDIM in all step-skipping conditions without quality compromise. The result is reported on CelebA.

|  |  | T=10 | T=50 | T= 100 | T=1000 |
|---|---|---|---|---|---|
| FID | DDIM | 28.72 | 9.385 | 7.912 | 6.755 |
|  | +DDSM | 28.33 | 8.826 | 7.132 | 6.039 |
| Total TFLOPs | DDIM | 0.498 | 2.494 | 4.988 | 49.88 |
|  | +DDSM | 0.260 | 1.376 | 2.384 | 19.58 |

We also present our DDSM can be combining with the famous Latent Diffusion Model (LDM), and further accomplish an excellent acceleration. For simplicity, we directly employ Stable Diffusion's encoder to obtain the latent vectors. The result is shown in Table 5. It shows that our DDSM combine well with the LDM on the CelebA dataset.

## 5.5 ANALYSIS OF THE SEARCHED RESULTS

Table 5: Combining DDSM with Latent Diffusion Model(LDM). The experiment is conducted on the CelebA dataset.

| Method | FID | avg. GFLOPs |
|---|---|---|
| LDM | 10.91 | 5.52 |
| LDM + DDSM | 10.76 | 2.22 |

In this section, we present the DDSM searched strategies using smoothed color bars and line graphs. The result is shown in Figure 4 (a) and (b). In the color bars, yellow denotes the smallest model, while green denotes the largest model. Middle-sized models between these two are presented with an interpolation of yellow and green.

For CIFAR-10 and ImageNet, DDSM generally tends to use larger models when the step number goes to zero. However, for CelebA, DDSM has a completely different tendency. It employs larger models at the beginning steps and leans towards smaller models as the step number decreases.

We attribute this phenomenon to the difference in the dataset attributes. In the beginning steps of generation, diffusion models are responsible for **coarse structure generation** (Choi et al., 2022). CIFAR-10 and ImageNet are multi-classes datasets of natural images with large variety. Due to the image diversity, various structures are included in the dataset, so the restriction of generating coarse structures of these images is very loose. In this case, the early-stage diffusion model can give any type of coarse structure, just as it is in Figure 4 (b). No matter what structure it generates, it can always find a proper object suitable for this structure. All it needs to do is to produce some inspiration for the latter generation. Therefore, the coarse structure generation task is easy in CIFAR-10 and ImageNet, resulting in small model usage in the DDSM's search result. On the other hand,

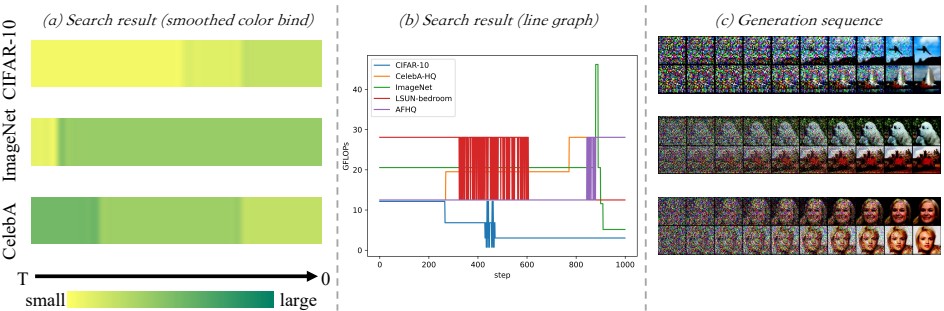

Figure 4: (a) The search results are presented in color bars. The yellow color denotes small models. The green color denotes large models. (b) The search results are presented in line graphs. (c) The generation sequences of CIFAR-10 and CelebA

in CelebA, the coarse structure of images is relatively uniform. Every image of CelebA should have an oval-like contour in the center, and the basic position of organs like eyes, nose, ears, and mouth should be fixed at some particular points, as shown in Figure 4 (b). Once the coarse structure goes wrong, the latter steps of the diffusion model can rarely fix this mistake. This restriction forces the diffusion model to possess a stronger structure-caption ability in the early stage, resulting in large model usage in the DDSM's search result.

In the latter steps of generation, diffusion models take charge of **detail generation**. For CIFAR-10 and ImageNet, every detail matters when considering image quality. When generating cars, wheels must be in the right shape and color. When generating birds, the beak and wings must have a correct pattern. However, in CelebA, these pixel-level details on faces do not largely affect the general image quality. Under the large resolution generation, the organs and structures have already been generated in the previous steps. Detail generation can be viewed as a simple interpolation of colors in CelebA generation.

In appendix, the search outcomes of LSUN-bedroom and AFHQ further prove our hypothesis. The LSUN-bedroom results paralleled those of CIFAR-10 and ImageNet, due to their shared high variety, while AFHQ echoed the patterns seen in CelebA-HQ, as both centrally feature face contours.

Table 6: FID scores of swapping the searched strategy in CIFAR10 and CelebA. Different rows denote different strategies. Different columns denote different datasets.

| Strategy \ Dataset | CelebA | CIFAR-10 |
|---|---|---|
| CelebA | **6.039** | 8.140 |
| CIFAR-10 | 7.431 | **3.552** |

We also conduct a strategy swap on these two datasets. The result is in Table 6. Applying CIFAR-10's strategy to CelebA leads to a great drop in performance, and vice versa, indicating the significance of searching for an optimal strategy for each dataset.

Overall, the step's importance distribution is highly dependent on the dataset. We must consider the influence of the dataset when trying to design a step-aware strategy. For concrete analysis of the behaviors of different steps, more theoretical proof and empirical efforts are required. We leave them for future work. We believe our method can be used as a good tool to find the importance of different steps. It may assist us in analyzing the in-depth theory of the responsibility of each step in the future.

## 6 CONCLUSION

In this paper, we re-examine the data generation steps in DDPMs and observe that the whole-network computation in all steps might be unnecessary and inefficient. For acceleration, we propose the step-aware network, DDSMs, as the solution. In DDSMs, we adopt the neural network in various sizes at different steps, according to generation difficulties. We conduct extensive experiments and show its effectiveness on CelebA, CIFAR-10, LSUN-bedroom, AFHQ, and ImageNet. DDSMs can present high-quality samples, with at most 76% computational cost saving.

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

APPENDIX

## A    EXPERIMENT ON LSUN-BEDROOM AND AFHQ

We've validated DDSM's effectiveness and scalability on two extra datasets, LSUN-bedroom (64x64) and AFHQ (64x64). LSUN-bedroom is entirely non-object-centric. While AFHQ is object-centric but from a domain different from CelebA. SaDiffusion achieves 59% and 71% acceleration on the respective datasets without compromising image quality, as detailed in Table 7.

The search results in Figure 5 of LSUN-bedroom and AFHQ further validate our analysis of the dataset's attributes. The LSUN-bedroom results parallels those of CIFAR-10, due to their shared high variety, while AFHQ echoed the patterns seen in CelebA-HQ, as both centrally feature face contours.

Table 7: Evaluation on LSUN-bedroom and AFHQ

| Method | LSUN-bedroom | | AFHQ | |
|---|---|---|---|---|
| | FID | GFLOPs | FID | GFLOPs |
| ADM | **5.027** | 49.88 | 7.244 | 49.88 |
| DDSM | 5.289 | **20.48** | **6.249** | **14.72** |

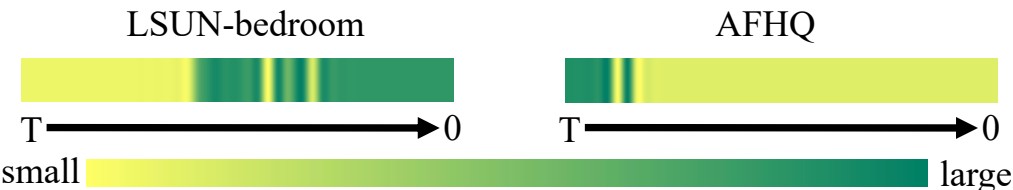

Figure 5: search results on LSUN-bedroom and AFHQ

## B    COMPARISON WITH CONCURRENT WORK DIFF-PRUNING

We elucidate the differences and provide an empirical comparison with the recent Diff-pruning Fang et al. (2023a;b). Diff-pruning employs a Taylor expansion over pruned timesteps to effectively reduce computational overhead. This approach primarily focuses on pruning the entire network. Our method, in contrast, adopts a strategy of adaptively pruning the network at various timesteps. This approach allows for a more nuanced and efficient solution. For a clearer understanding of these differences, we conducted an empirical comparison with Diff-pruning. Table 8 showcases our experiment on CelebA 64x64 using 100 steps. Our DDSM not only achieves a FID comparable to Diff-pruning but also demonstrates superior efficiency.

Table 8: Experimental comparison with Diff-Pruning, on CelebA 64x64, 100 steps.

| Metric | Baseline | Diff-pruning | DDSM |
|---|---|---|---|
| FID (Lower is better) | 6.48 | 6.24 | **6.04** |
| GFLOPS (Lower is better) | 49.9 | 26.6 | **19.6** |

## C    COMBINING WITH THE EDM SAMPLER

Our DDSM is compatible with recent sampling schedulers, such as DPM Lu et al. (2022a), DPM++ Lu et al. (2022b), uniPC Zhao et al. (2023), and EDM Karras et al. (2022). These samplers are primarily focused on devising novel noise schedules to enhance the performance of diffusion models. Our method, which adaptively prunes at different steps, is theoretically compatible with these approaches. Among these samplers, we have chosen EDM for our experiments, as we believe

this selection could demonstrate our method's suitability for similar methods. In the experiment, we kept the pretrained weights of DDSM. We then replaced the original DDPM scheduler with EDM's Heun Discrete 2nd order method and initiated a new search process. The results, as illustrated in Table 9, show that DDSM effectively integrates with EDM on the unconditional CIFAR-10 dataset, achieving 45% and 56% total TFLOPs saving for 50 steps and 100 steps. Note that, the quality can be further improved when retraining the network with EDM's setting.

Table 9: Experiment result of combining DDSM with EDM, on CIFAR-10.

| Metric | EDM50 | EDM50+DDSM | EDM100 | EDM100+DDSM |
|---|---|---|---|---|
| FID | 3.65 | 3.61 | 3.41 | 3.49 |
| Total TFLOPs | 0.61 | 0.34 | 1.22 | 0.54 |

## D  EXPERIMENT ON CELEBA-HQ 128x128

We conduct experiments on CelebA-HQ-128x128 to prove that our DDSM is applicable with higher resolution data. In this experiment, we train a new slimmable supernet from scratch, and directly employ the search result on CelebA-HQ-64x64 to super network. Table 10 shows the result.

Table 10: Experiment result on CelebA-HQ 128x128.

| Metric | ADM | DDSM |
|---|---|---|
| FID (Lower is better) | 7.53 | 7.71 |
| GFLOPS (Lower is better) | 194.00 | 76.18 |

## E  EXPERIMENT ON CONDITIONAL CIFAR-10

We also conduct experiments on conditional CIFAR-10 to prove that our DDSM is applicable with guided diffusion. In this experiment, we train a guided diffusion slimmable supernet from scratch, and directly employ the search result on unconditional CIFAR-10 to super network. Table 11 shows the result.

Table 11: Experiment result on conditional CIFAR-10.

| Metric | ADM | DDSM |
|---|---|---|
| FID (Lower is better) | 2.48 | 2.52 |
| GFLOPS (Lower is better) | 12.14 | 6.20 |

## F  IMPLEMENTATION DETAILS

**Architecture**  Our implementation of slimmable networks draws inspiration from the US-Net. These networks are characterized by their ability to operate at various widths, providing a flexible and universal solution for network scalability. During the training of slimmable networks, we focus on optimizing the smallest, largest, and a selection of randomly sampled middle-sized sub-networks. This approach implicitly enhances the performance of all potential networks within the supernet.

In practical terms, our slimmable UNet adheres to the structure of ADM, but with a significant modification: all convolution layers are replaced by slimmable convolutions. These specialized convolutions are capable of adaptively processing tensors with a varying number of input channels. To accommodate a broader range of sub-networks, we adjusted the group number in the group normalization layer from 32 to 16. Additionally, we chose to omit the Batch Normalization (BN) calibration stage, as proposed in [11], since our diffusion UNet exclusively utilizes GroupNorm.

Regarding the sizes of the sub-networks, we offer seven different options, corresponding to $\frac{2}{8}$, $\frac{3}{8}$, $\frac{4}{8}$, $\frac{5}{8}$, $\frac{6}{8}$, $\frac{7}{8}$, and $\frac{8}{8}$ of the original ADM's width. To find more efficient strategies, we manually exclude

some large width options. For example, in CIFAR-10, we exclude the $\frac{7}{8}$ and $\frac{8}{8}$ options. In most cases, the total strategy space is of $7^{num\_timesteps}$.

**Search and Evaluation**   In the searching phase, the FID was computed by comparing the entire training dataset with the generated images. Empirical findings indicate that the FID score offers more consistent performance measurement compared to the loss; therefore, we aimed to search for a strategy yielding a lower FID. For the search parameters, the process encompasses a total of 10 iterations, with each iteration involving a population of 50, and maintaining a mutation rate of 0.001. The initial generation crucially includes a mix of uniform non-step-aware strategies and some random strategies. This specific approach to initialization and mutation has been empirically found to facilitate easier convergence of the search algorithm. Furthermore, a weight parameter is incorporated, which multiplies the GFLOPs to strike a balance between image quality and computational efficiency. For CIFAR-10, we set the weight parameter to 0.1 to favor higher image quality, while for CelebA, the FLOPs weight parameter is adjusted to 0.25. These parameters were manually selected to ensure that there are no compromises in generation quality.

**Compatibility Experiments**   For the DDIM experiment, we adopted the weights from the 1000-step ADM model and applied the DDIM sampling schedule. As indicated in Table 4, our method significantly boosts DDIM's speed by 48%, 45%, 53%, and 62% for 10, 50, 100, and 1000 steps, respectively. This suggests that while additional steps enhance generation quality, they also introduce excess computational load. Our approach proves increasingly beneficial as the number of steps increases. In our latent diffusion experiment, we employed the AutoEncoderKL of SD1.4 to transform images into latent vectors, upon which our DDSM was trained. To adapt to the reduced spatial size, we modified the U-Net downsampling from 4 to 2. Our DDSM achieved a 60% acceleration in latent diffusion methods.

## G   DISCUSSION ON THE EXTRA COST

In this section, we discuss the extra cost introduced by our DDSM in the training and searching stage. Although we introduce extra training and search cost, DDSM still shows its advantages in inference. In deep learning algorithms, a common manner is we train a model once and repeatedly use this model to infer results. Nowadays, well-known Diffusion models all conform to this manner, like StableDiffusion and Imagen. They require a large amount of time to train. But once they are trained, they could repeatedly generate billions of images. Therefore, in terms of efficiency, we usually care more about the inference speed due to its repetitiveness.

Our method aims to reduce the inference cost of diffusion models. We prune the diffusion inference process with its step-aware strategy, accomplishing at most 76% acceleration. These cost savings could be counted repeatedly. The more samples we produce, the more benefits our framework demonstrates. In DDSM, we indeed introduce extra training and search cost, but these costs only take once. Concretely, the training cost of the slimmable network is about 2 times to 3 times of the normal diffusion model's training. The search cost depends on the search hyper-parameters. In our setting, it is approximately the same time as training a normal diffusion model. In Table 12, we present the time cost of our DDSM on NVIDIA RTX 3090's GPU hours.

Table 12: Total GPU hours of our DDSM.

|  | CIFAR-10 | CelebA |
|---|---|---|
| DDPM-train | 278 | 435 |
| DDSM-train | 502 | 1036 |
| DDSM-search | 320 | 524 |

## H   SEPARATED LINE GRAPHS OF SEARCH RESULTS

We plot the search result separately for better visualization, as shown in Figure 6

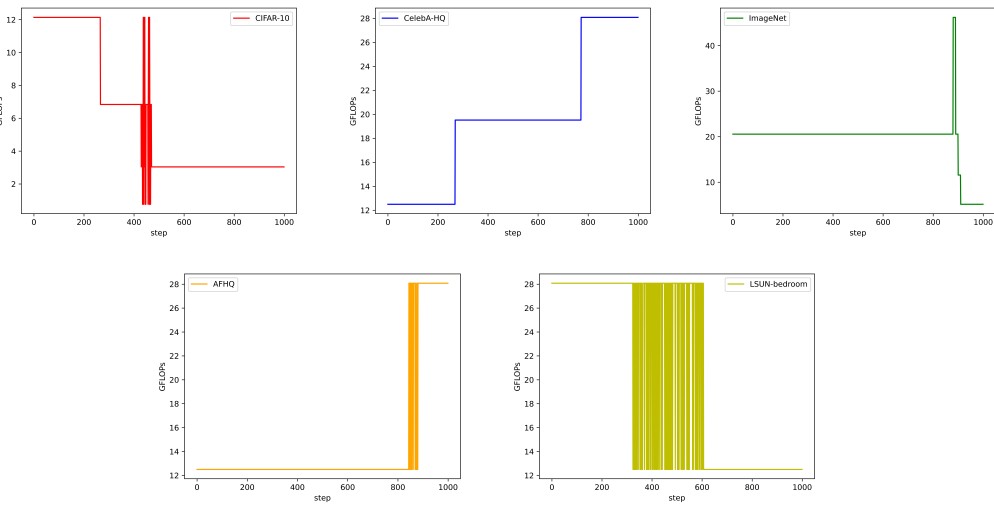

Figure 6: Line graphs of the search results across all 5 datasets.

# I QUALITATIVE RESULT

We present the generation result of DDSM in Figure 7. It shows that the efficient DDSM could produce high-quality images.

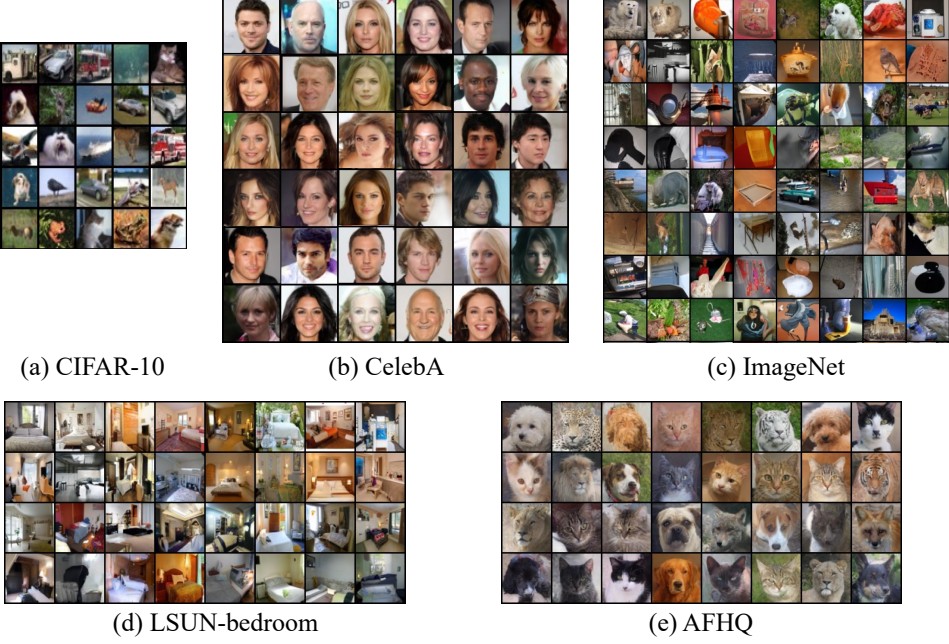

Figure 7: Generated images of DDSM on CIFAR-10, CelebA, ImageNet, LSUN-bedroom, and AFHQ.

