# OpenReview forum: "Denoising Diffusion Step-aware Models"
_ICLR.cc/2024/Conference — ICLR 2024 poster_

### Official Review · Reviewer_buDR · 2023-10-17

**Soundness:** 3 good
**Presentation:** 3 good
**Contribution:** 3 good
**Rating:** 8
**Confidence:** 4

**Summary:**

This paper presents a novel approach to accelerating sampling from denoising diffusion models without sacrificing image quality. This approach is based on the idea that not every generation step is equally difficult nor equally important for the quality of the final image. The authors propose using *slimmable networks* to prune the network weights at less-important generation steps, and combine this with an evolutionary search method to identify which steps can be safely pruned without sacrificing image quality. The authors demonstrate that their method reduces the FLOPs required for multiple datasets while achieving comparable or better sample quality (FID) than the unpruned methods, and further show that it can be combined with the adaptive step-skipping of DDIM or with latent diffusion for further improvements. They also visualize the learned pruning schedules for different datasets, and show that the set of safely-prunable generation steps differs depending on dataset structure.

**Strengths:**

**[S1]** The idea of pruning the network adaptively based on the difficult of each generation step is insightful, and well-motivated by the introduction and the pilot study in section 3.1. This approach also seems quite novel. I'm not aware of any previous work that has considered adaptively using different network sizes at different timesteps.

**[S2]** The authors show that their approach is actually complementary to existing acceleration methods like DDIM, and yields additional improvements when combined. I expect that the approach could also be combined with some of the more recent ODE-based acceleration methods like DPM-Solver, since the proposed DDSM technique is agnostic to the exact mathematical sampling process and instead adaptively shrinks the network itself.

**[S3]** The empirical results are quite strong across multiple datasets. The authors show significant cost savings relative to sampling from the full model, and actually show slight *improvements* in sample quality for all but one of the datasets (LSUN-bedroom takes a small hit).

**[S4]** The approach is easy to understand and seems like it would not be too difficult to implement. The authors also plan to release their code.

**[S5]** I found the results of the pruning schedule search to be quite insightful. Different datasets show different amounts of pruning, and this pruning occurs at different times, which seems to match whether the dataset has more information in its high-frequency or low-frequency components.

**Weaknesses:**

**[W1]** The approach relies on "slimmable networks" and "slimmable counterparts" of standard convolution and normalization. However, these aren't discussed in much detail. I think the paper could benefit from some additional background on what slimmable networks are and how they work, and more details on the particular slimmable network architecture used for this work. (In particular, how many slimmable switches were used, and how do they fit into the U-Net architecture?)

**[W2]** The objective for the evolutionary search algorithm is also presented at a fairly high level and could use more details. The authors mention using NSGA-II to "balance conflicting objectives", but it's not clear to me what objectives were used. Algorithm 2 suggests that the search objective was a linear combination of FID and FLOPS, but the details of the linear combination weights are not specified.

**[W3]** Although the contributions of this work do seem orthogonal to some of the more recent work in accelerating diffusion model sampling, I think the experimental results would be more impressive if they could also be demonstrated for these more recent accelerated sampling approaches like DPM-Solver, [DPM-Solver++](https://arxiv.org/abs/2211.01095) or the recent [UniPC](https://arxiv.org/abs/2302.04867). My guess is that the methods could be combined, but it would be useful to see how much FLOPs can be saved when combined with these more recent samplers.

**Questions:**

Could you provide more details on the slimmable network architecture and on the configuration of the search algorithm, as I discuss in [W1] and [W2]? For the search algorithm, what was the value of $w_M$ used, and how critical is this choice? (Or, are the FID and FLOPs measurements automatically balanced by NGSA-II somehow?)

Figure 4a and Figure A are quite interesting, but they also look somewhat blurry. I can sort of see some fuzzy "bands" at different points of the trajectory; are these actual changes in the slimmed model size or are these some sort of compression artifact in the figure image? Also, what is the resolution of the X axis here, e.g. where are the boundaries between different steps? It's hard to understand exactly what the plot is showing, and I think this plot would be more readable if it were presented as a line graph (instead of as a heatmap), since its only 1 dimensional.

I found some very recent related work ["Structural Pruning for Diffusion Models" (Fang et al. 2023)](https://arxiv.org/abs/2305.10924) which was just accepted at NeurIPS. That work also considers pruning in order to speed up diffusion model inference, although I believe they only consider pruning the entire network rather than adaptively pruning at different timesteps. It might make sense to discuss this related work in your paper, and I'd also be curious how your approach compares to theirs in terms of FLOPs savings.

Have you explored combining your DDSM approach with some of the more recent accelerated sampling approaches? And do you still observe runtime improvements for those methods?

---

> ### Author Response · Authors · 2023-11-19
>
> **Background of slimmable networks**
>
> Our implementation of slimmable networks draws inspiration from the US-Net [15]. These networks are characterized by their ability to operate at various widths, providing a flexible and universal solution for network scalability. During the training of slimmable networks, we focus on optimizing the smallest, largest, and a selection of randomly sampled middle-sized sub-networks. This approach implicitly enhances the performance of all potential networks within the supernet.
>
> In practical terms, our slimmable UNet adheres to the structure of ADM, but with a significant modification: all convolution layers are replaced by slimmable convolutions. These specialized convolutions are capable of adaptively processing tensors with a varying number of input channels. To accommodate a broader range of sub-networks, we adjusted the group number in the group normalization layer from 32 to 16. Additionally, we chose to omit the Batch Normalization (BN) calibration stage, as proposed in [11], since our diffusion UNet exclusively utilizes GroupNorm. Regarding the sizes of the sub-networks, we offer seven different options, corresponding to 2/8, 3/8, 4/8, 5/8, 6/8, 7/8, and 8/8 of the original ADM's width. This results in a total strategy space of 7^(num_timesteps).
>
> **Details of the search algorithms**
>
> For the search parameters, the process encompasses a total of 10 iterations, with each iteration involving a population of 50, and maintaining a mutation rate of 0.001. The initial generation crucially includes a mix of uniform non-step-aware strategies and some random strategies. This specific approach to initialization and mutation has been empirically found to facilitate easier convergence of the search algorithm. Furthermore, a weight parameter is incorporated, which multiplies the GFLOPs to strike a balance between image quality and computational efficiency. For CIFAR-10, we set the weight parameter to 0.1 to favor higher image quality, while for CelebA, the FLOPs weight parameter is adjusted to 0.25. These parameters were manually selected to ensure that there are no compromises in generation quality.
>
> **Elaboration on the search result**
>
> For visual propose, I smoothed the color binds to present a general trend instead of the details. The fuzzy area are some mutation points of the search result. They are actual changes in model size. I think your suggestion of drawing line graph is very useful. Therefore, I plot them in our modified version in the **Appendix Section D**.
>
> **Discussion on “Structural Pruning for Diffusion Models”**
>
> Please see the comment to all reviewers.
>
> **Compatible with recent fast solvers**
>
> Please see the comment to all reviewers.
>
>
> Finally, I sincerely appreciate the valuable insights provided by the reviewers and warmly welcome any further criticisms or suggestions. Your timely feedback is crucial for the enhancement of this paper. I look forward to your valuable comments.
>
>
> [15] Universally Slimmable Networks and Improved Training Techniques, Yu et al..

---

> > ### Comment · Reviewer_buDR · 2023-11-20
> > **Discussion**
> >
> > **Details on slimmable networks and search algorithms:** Thanks for the additional details. I think you should also add these details to the paper so that your results can be reproduced, it doesn't look like they are currently present.
> >
> > **Visualization of search results:** I see. I don't think you should smooth the color bands in Figure 4, since that does not accurately represent what your method does. The plots in the appendix are much more readable and I would suggest replacing the color band visualizations with these line plots in the main paper.
> >
> > I am surprised that your search results have such sudden jumps back and forth between sizes. It seems unlikely to me that this represents the best schedule. How much variability did you observe between the search results over different search attempts?
> >
> > **Diff-Pruning / EDM results:** Thanks for providing these new results. It doesn't look like these results have actually been added to the paper yet, are you planning to add them?

---

> > > ### Author Response · Authors · 2023-11-21
> > >
> > > Thanks for your remind. In this revised version, we have added experiments, discussion, and detailed content to the original paper. The newly added and modified content is highlighted in red for easy identification. Additionally, we have included the line graph in the main content of our paper for clearer illustration.
> > >
> > > Regarding the observed fluctuations in our search results, we attribute these to uncertain timesteps of the diffusion process. In multiple CIFAR-10 experiment runs, the fluctuation area consistently appears within the 400-600 step range, and the internal patterns of these fluctuation areas show little similarity across different runs. Despite this, the overall trend in all the searched strategies demonstrates a similar behavior of increased reliance on larger models as the timestep approaches T→0.
> > >
> > > Based on these observations, we hypothesize that: 1) there are some uncertain timesteps that are challenging for our search algorithm to accurately pinpoint, and 2) our current search strategy may not have reached an 'optimal' point. Nevertheless, this does not largely weaken our method, since our results have clearly demonstrated that our approach is significantly more effective than the original non-step-aware diffusion. Moreover, pursuing an extremely optimal result would entail substantially higher computational costs, we believe that our current strategy has achieved a favorable balance between efficiency and performance.

---

### Official Review · Reviewer_X3mN · 2023-10-19

**Soundness:** 3 good
**Presentation:** 2 fair
**Contribution:** 2 fair
**Rating:** 6
**Confidence:** 5

**Summary:**

The paper proposes Denoising Diffusion Step-aware Models (DDSM) to improve efficiency of denoising diffusion probabilistic models (DDPMs) for image generation. Previously, DDPMs require compute-intensive iterative sampling, using the full model each step. DDSM hypothesizes different steps have varying importance, and uses a spectrum of networks with adapted sizes for each step, determined via evolutionary search. This avoids redundant computation in less critical steps.
DDSM integrates with slimmable networks - trained simultaneously on sub-networks to enable execution at arbitrary sizes.

**Strengths:**

DDSM accelerates diffusion models by avoiding uniform computation for all steps. The step-aware network design is shown to be efficient and effective across datasets.

**Weaknesses:**

The major concern is the pathway of denoising acceleration.  It's evident that the prevalent approach to enhancing diffusion speed, in the context of network compression, hinges on the application of post-training quantization techniques. These techniques enable the compression of neural networks in a manner that circumvents additional training [1,2]. However, I observe that your method necessitates training and, notably, falls short in performance when compared to methods that forego training. To illustrate, empirical assessments show that DDSM facilitates computational reductions of 49% for CIFAR-10 and 76% for ImageNet. In contrast, the techniques in [1,2] manage to achieve 4 or 8-bit quantization (surpassing DDSM in speed) without compromising the FID score. It's crucial to underscore that these methods achieve this efficiency entirely without the need for further training. Consequently, the approach employed by DDSM for accelerating denoising doesn't appear to be robust enough.


[1] PTQ4DM, CVPR 2023

[2] Q-dfiifusion, ICCV 2023

**Questions:**

Refer to weaknesses.

---

> ### Author Response · Authors · 2023-11-19
>
> **Model Compression Techniques: Pruning and Quantization**
>
> Thank you for your insightful comments and constructive criticism regarding our paper. We appreciate the opportunity to clarify and further elaborate on our work, especially in relation to the concerns raised about the pathway of denoising acceleration and the comparison with post-training quantization techniques. In the field of model compression, discerning the roles of model pruning and weight quantization is crucial, as these are two distinct yet synergistic approaches to accelerating model performance. Model pruning is primarily concerned with eliminating superfluous weights in a network. Conversely, quantization involves converting the data type of weights and activations into lower precision formats, such as int8 or int4, to reduce computational demands.
>
> Although these methods are individually powerful, their combined application, typically applied sequentially on a pretrained model, can further enhance the performance. For comprehensive insights, [10] presents a detailed survey on the collaborative use of these techniques for optimal model efficiency. Additionally, [11] and [12] showcase practical implementations where both weight pruning and quantization are simultaneously applied to a model.
>
> It's noteworthy that pruning typically necessitates additional training or fine-tuning, as opposed to quantization which generally only requires calibration. However, the unique contributions of pruning, due to its orthogonal nature to quantization, should not be underestimated.
>
> **Distinctive Aspects of DDSM and Quantization-Based Methods**
>
> We would like to emphasize the unique aspects of DDSM when compared to the quantization-based methods mentioned, such as PTQ4DM [13] and Q-diffusion [14]. The key point of our approach is the step-awareness in diffusion models. Unlike these methods, which apply a uniform model across all steps, DDSM tailors a spectrum of network sizes adapted for each diffusion step. This is a significant departure from PTQ4DM and Q-diffusion’s approach, which leads to a uniformly quantized model for all steps.
>
> In conclusion, we acknowledge the points raised and appreciate the comparison with other methods and we will discuss it in our modified version. We believe that DDSM contributes a unique perspective to the field of diffusion model acceleration. We also see potential for future work, possibly exploring how DDSM might be combined with quantization techniques to further enhance its performance. I sincerely appreciate the valuable insights provided by the reviewers and warmly welcome any further criticisms or suggestions. Your timely feedback is crucial for the enhancement of this paper. I look forward to your valuable comments.
>
> [10] Pruning and Quantization for Deep Neural Network Acceleration: A Survey, Liang et al., *Elsevier.*
>
> [11] PQK: Model Compression via Pruning, Quantization, and Knowledge Distillation, Kim et al., INTERSPEECH 2021.
>
> [12] Once Quantization-Aware Training: High Performance Extremely Low-Bit Architecture Search, Shen et al., ICCV2021.
>
> [13] Post-training Quantization on Diffusion Models, Shang et al., CVPR 2023.
>
> [14] Q-Diffusion: Quantizing Diffusion Models, Li et al., ICCV 2023.

---

> > ### Comment · Area_Chair_cg3y · 2023-11-20
> > **Please respond to the authors' rebuttal**
> >
> > Dear reviewer,
> >
> > The window for interacting with authors on their rebuttal is closing on Wednesday (Nov 21st). Please respond to the authors' rebuttal as soon as possible, so that you can discuss any agreements or disagreements. Please acknowledge that you have read the authors' comments, and explain why their rebuttal does or does not change your opinion and score.
> >
> > Many thanks,
> >
> > Your AC

---

> ### Comment · Reviewer_X3mN · 2023-11-21
> **Response to Authors**
>
> I appreciate the authors' efforts in discussing the distinctions between quantization and pruning. However, I'd like to **address some misconceptions regarding quantization in the authors' responses**. The authors state that "pruning typically necessitates additional training or fine-tuning, as opposed to quantization which generally only requires calibration". This statement overlooks the complexity of quantization techniques. Specifically, quantization can be divided into two main categories: post-training quantization and quantization-aware training. The latter, contrary to the statement, does require training. For instance, the method used in "SnapFusion: Text-to-Image Diffusion Model on Mobile Devices within Two Seconds" (presented at NeurIPS 2023, with the code and paper released in July) is a clear example of a quantization-aware training approach for speeding up diffusion models.
>
> Additionally, the authors' claim that quantization and pruning are orthogonal techniques warrants further scrutiny. There is substantial research suggesting that due to the inherent trade-off between a network's representational capacity and its degree of over-parameterization, quantization and pruning might not be effectively employed simultaneously. This aspect of interaction between the two techniques seems to be overlooked in the current discussion.
>
> Given these considerations, I believe that the role of quantization, particularly in diffusion acceleration tasks, should be revisited. Its simplicity and straightforward application make it a well-studied technique to consider in the context of the discussed research.
> Overall, I will not change my score.

---

> > ### Comment · Area_Chair_cg3y · 2023-11-21
> > **Please provide references**
> >
> > Dear reviewer X3mN,
> >
> > Thank you for replying to the authors. Can you please provide references that refer to the "substantial research" in the following the statement: "There is substantial research suggesting that due to the inherent trade-off between a network's representational capacity and its degree of over-parameterization, quantization and pruning might not be effectively employed simultaneously".
> >
> > Many thanks,
> >
> > Your AC

---

> > > ### Comment · Reviewer_X3mN · 2023-11-21
> > > **Quantization and pruning are not simply orthogonal.**
> > >
> > > The pruning-quantization strategy is more complex than either pruning or quantization alone. In other words, it is almost impossible for manually tuning the pruning ratios and quantization codebooks at fine-grained levels [1]. To address this issue, recent methods adopt certain optimization techniques to optimize this problem. Thus, quantization and pruning are not simply orthogonal as the authors claimed. If one wants to realize quantization and pruning simultaneously, one needs to optimize this problem very carefully.
> > >
> > >
> > > OPQ: Compressing Deep Neural Networks with One-shot Pruning-Quantization, AAAI 2021

---

> > > > ### Author Response · Authors · 2023-11-21
> > > >
> > > > We are glad to solve our reviewer’s concern.
> > > >
> > > > **While quantization is indeed valuable, it does not diminish the importance of our pruning work.** It's crucial to note that quantization requires specific hardware compatibility to function effectively. Deployment through ONNX and TensorRT is necessary, and the performance benefits of quantization also depend on the type of GPU or CPU used. In many cases, without appropriate hardware, the acceleration gains are merely simulated and not actual. Additionally, models post-quantization, if subjected to further fine-tuning, must revert to the fp32 format, negating the acceleration benefits during this fine-tuning phase. In contrast, pruned models are inherently fine-tune friendly, maintaining their speed advantage even during fine-tuning. Thus, the value of pruning work stands firm, unaffected by the quantization process.
> > > >
> > > > **The paper referred to by the reviewer focuses on image classification, a domain distinctly separate from the insights of pruning in diffusion models.** This distinction is critical as it implies that the reviewer's claim may lack relevance to our work. The field of diffusion models operates under different parameters and challenges, making a direct comparison with image classification models less effective in undermining the significance of pruning in our context.
> > > >
> > > > **Numerous pruning works exist across various areas, demonstrating utility independent of quantization.** These works [1, 2, 3, 4, 5, 6, 7, 8, 9, 10, 11] highlight that pruning can be highly beneficial in its own right, without necessarily being coupled with quantization. These papers, encompassing a diverse range of domains including image classification, object detection, semantic segmentation, large language models, image compression and image generation, underscore that the integration of quantization is not a ubiquitous practice in the field. In our specific work, pruning stands as a distinct and out-of-scope area, separate from the considerations of quantization.
> > > >
> > > > **Rethinking Model Compression Discussions:**
> > > > In conclusion, we encourage a reconsideration of a pivotal question: "**Is it necessary to discuss quantization in every pruning paper?**" Based on our explanations above and the standard practices observed in other works, we believe the answer is no. It's essential to recognize the unique contributions and contexts of different compression techniques, acknowledging that each has its place and significance independent of the others.
> > > >
> > > >
> > > > 1. Learning to Jointly Share and Prune Weights for Grounding Based Vision and Language Models. (2023). In Proceedings of the International Conference on Learning Representations (ICLR).
> > > > 2. Pruning Deep Neural Networks from a Sparsity Perspective. (2023). In Proceedings of the International Conference on Learning Representations (ICLR).
> > > > 3. Prune Spatio-temporal Tokens by Semantic-aware Temporal Accumulation. (2023). In Proceedings of the International Conference on Computer Vision (ICCV).
> > > > 4. Structural Alignment for Network Pruning through Partial Regularization. (2023). In Proceedings of the International Conference on Computer Vision (ICCV).
> > > > 5. Differentiable Transportation Pruning. (2023). In Proceedings of the International Conference on Computer Vision (ICCV).
> > > > 6. Learning Dynamic Routing for Semantic Segmentation (2020). In Proceedings of the Computer Vision and Pattern Recognition (CVPR).
> > > > 7. LLM-Pruner: On the Structural Pruning of Large Language Models. (2023). In Proceedings of the Neural Information Processing Systems (NeurIPS).
> > > > 8. Sheared LLaMA: Accelerating Language Model Pre-training via Structured Pruning. 2023. Preprint.
> > > > 9. A Simple and Effective Pruning Approach for Large Language Models. (2023). In Proceedings of the International Conference on Machine Learning (ICML) Workshop.
> > > > 10. AdaNIC: Towards Practical Neural Image Compression via Dynamic Transform Routing. (2023). In Proceedings of the International Conference on Computer Vision (ICCV).
> > > > 11. Structural Pruning for Diffusion Models, Fang et al., (2023). In Proceedings of the Neural Information Processing Systems (NeurIPS).

---

> > > > > ### Comment · Reviewer_X3mN · 2023-11-23
> > > > >
> > > > > The authors addressed most of my concerns. I raised my score to 6.

---

### Official Review · Reviewer_339s · 2023-10-30

**Soundness:** 2 fair
**Presentation:** 2 fair
**Contribution:** 2 fair
**Rating:** 6
**Confidence:** 4

**Summary:**

This paper...
- proposes to accelerate diffusion sampling by using diffusion models of different size at each time-step,
- proposes to use evolutionary search to find a best step-aware strategy,
- shows the effectiveness of the proposed approach on CIFAR-10, CelebA-HQ, and ImageNet sampling.

**Strengths:**

- Paper is easy to follow.
- Using networks of different size (slimmable network) for each-time step is an unexplored approach in diffusion acceleration.
- DDSM shows promising performance gains on a variety of generation tasks.

**Weaknesses:**

While the idea behind DDSM is interesting, I am inclined to give "marginal reject" due to weak experimental validation.

- The paper lacks comparison with [1], which I think is a very relevant work.
- The paper lacks experiments on higher resolution data. Can the authors provide results on $\geq 128$ resolution images?

[1] Structural Pruning for Diffusion Models, Fang et al., NeurIPS, 2023.

**Questions:**

- Figure 3 is unclear. What do networks with mixed colors (blue and green) mean?
- In Table 3, why do DDSM outperform ADM-large? Shouldn't the performance of DDSM be bounded by the performance of the largest model?
- Is DDSM compatible with guided diffusion? Can the authors provide some demonstrations?
- Is DDSM compatible with recent fast solvers, such as EDM [2]?

[2] Elucidating the Design Space of Diffusion-Based Generative Models, Karras et al., NeurIPS, 2022.

---

> ### Author Response · Authors · 2023-11-19
>
> **Discussion on “Structural Pruning for Diffusion Models”**
>
> Please see the comment to all reviewers.
>
> **Compatible with recent fast solvers**
>
> Please see the comment to all reviewers.
>
> **Explanation of Figure 3**
>
> We apologize for any confusion caused and have now provided a more detailed description in the caption of Figure 3 for clarity. In this figure, the color green represents the activated weights within the network, whereas blue represents the inactive weights. The varying proportions of blue and green in different networks indicate networks of various sizes. The supernet, which has all of its weights activated, is depicted entirely in green.
>
> During the training phase, our focus is on the supernet, which we train by optimizing its various sub-networks. This step is crucial for establishing a foundation of diverse network sizes and capabilities. In the subsequent searching stage, we creatively combine these sub-networks into different step-aware strategies. The objective here is to explore and identify the most optimal strategy, one that effectively balances performance and efficiency.
>
> **Explanation of DDSM outperforming ADM-large**
>
> In Table 4, our DDSM model demonstrates a marginal improvement over ADM-large, reflected in a FID score enhancement of 0.161. We believe this improvement is largely due to the inherent randomness in the image generation process. To substantiate this claim, we conducted additional experiments using three distinct seeds for image regeneration. The results, detailed in the table below, reveal that the FID scores of DDSM are closely aligned with those of ADM-large, suggesting comparable performance between the two models.
>
> | Seed | ADM-Large FID | DDSM FID |
> | --- | --- | --- |
> | 1 | 3.713 | 3.552 |
> | 2 | 3.917 | 3.878 |
> | 3 | 3.332 | 3.639 |
> | Mean | 3.654 | 3.690 |
>
> DDSM's exceptional performance on ImageNet, surpassing ADM-large by a margin of 2.286, illustrates that pruned models can indeed outperform their unpruned counterparts. This is not a new concept and aligns with findings in prior research. For example, [7] improves 0.9 accuracy with 40% pruned DenseNet, [8] and [9] improves MobileNetV2 by 3.4 and 3.2 with less parameters. The success of DDSM is due to effective pruning strategies that eliminate unnecessary parameters, leading to a more efficient model focused on key features. Additionally, pruning acts as a regularizer that helps in enhancing the model's generalization ability, allowing it to perform better on diverse datasets like ImageNet.
>
> Finally, I sincerely appreciate the valuable insights provided by the reviewers and warmly welcome any further criticisms or suggestions. Your timely feedback is crucial for the enhancement of this paper. I look forward to your valuable comments.
>
> [7] "Learning Efficient Convolutional Networks through Network Slimming," Liu et al., ICCV, 2017.
>
> [8] "Once for All: Train One Network and Specialize it for Efficient Deployment," Cai et al., ICLR, 2020.
>
> [9] "AutoSlim: Towards One-Shot Architecture Search for Channel Numbers", Yu et al., NIPS workshop, 2019.
>
> **Compatibility with more diffusion settings**
>
> The compatibility of our DDSM is rooted in our core design: adaptively allocate computational resources in a step-aware manner. This step-awareness make our method orthogonal to these advanced applications.
>
> We recognize your concerns regarding the need for more experimental validation. To address this, we have initiated experiments with CelebA at a resolution of 128x128 and classifier-guided diffusion on CIFAR-10. These experiments are time-intensive, we are not sure whether we can obtain a result by the time of rebuttal due date. But we commit to sharing the results as soon as they are available.
>
> So far, our work has undergone validation across five different datasets in various domains. With the addition of these newly added experiments, our aim is to potentially broaden the scope of our research’s validation, hoping to encompass a wider array of applications.

---

> > ### Comment · Reviewer_339s · 2023-11-20
> >
> > I appreciate the authors' reply. While this work has interesting practical implications, there are no theoretical contributions. So, I believe I must apply a higher standard to experimental validation, and I think $64 \times 64$ resolution results are not sufficient for ICLR. I will keep my initial rating, and wait for $\geq 128 \times 128$ resolution results.

---

> > > ### Comment · Area_Chair_cg3y · 2023-11-20
> > > **Request to reviewer**
> > >
> > > Dear reviewer 339s,
> > >
> > > Thank you for replying to the authors' rebuttal. The authors seem to have addressed 1 out of the 2 weakness points that you raised. Given this, and taking into account the other reviews, can you explain why this is not enough to raise your score at all? If you think there is a clear need for  >=128x128 resolution experiments, can you explain why? For example, are you perhaps expecting that there are parts of the proposed method that do not to scale well, or are the authors making a claim on the scalability of the method? Stating that 64x64 resolution results are not sufficient for ICRL needs to be explained a little further, as there is no such general policy in place as far as I am aware.
> > >
> > > Many thanks,
> > >
> > > Your AC

---

> > > > ### Comment · Reviewer_339s · 2023-11-20
> > > >
> > > > I apologize for my vague remark. As this work is purely empirical, the significance of this work is decided by the practicality of DDSM. There are two reasons why I believe results on $\geq 128 \times 128$ resolution images are important in order to accurately gauge the practicality of DDSM.
> > > >
> > > > - First, compared to previous work on making diffusion models efficient such as pruning [1] / distillation [2] / better integrators [3] **which are applicable to any pre-trained diffusion models**, this paper requires **both the training of a new diffusion model and sampling strategy search through evolutionary search**. This makes this technique more expensive compared to [1,2,3], which naturally raises the question, is DDSM really worth using over works such as [1,2,3]? Specifically, it is well known that diffusion model training becomes extremely expensive (both in terms of computation and time) as image resolution increases. DDSM requires additional evolutionary search as well. Thus, for me to accurately compare the drawback of DDSM (training cost + evolutionary search cost) against the contribution of DDSM (sampling acceleration), I need additional results on $\geq 128 \times 128$ resolution images.
> > > >
> > > > - Second, previous works on efficient sampling of diffusion models [1,2,3] provide at least one result on $\geq 128 \times 128$ resolution data. For instance, on LSUN bedroom $256 \times 256$, [2] achieves 5.22 FID with 2 NFE, and  [3] achieves $\leq 3$ FID under 20 NFEs. For me to gauge the significance of this work compared to previous works [1,2,3] accurately under all settings, I need results on higher resolution data.
> > > >
> > > > [1] Structural Pruning for Diffusion Models, Fang et al., NeurIPS, 2023.
> > > >
> > > > [2] Consistency Models, Song et al., ICML, 2023.
> > > >
> > > > [3] DPM-Solver: A Fast ODE Solver for Diffusion Probabilistic Model Sampling in Around 10 Steps, Lu et al., NeurIPS, 2022.

---

> ### Author Response · Authors · 2023-11-21
>
> Hi. Thanks for your comments. We would like to update our empirical results on CelebA-HQ 128x128 and conditional CIFAR-10.
>
> Our experiments with CelebA-HQ at a resolution of 128x128 have demonstrated the efficacy of our DDSM with higher resolution data. In this study, we trained a new slimmable supernet for 128x128 resolution data from scratch. We then applied the strategy initially searched on the 64x64 CelebA-HQ dataset to this supernet. The results were promising: DDSM achieved a 61% reduction in FLOPs while preserving the quality of generation.
>
> Similarly, our experiments on conditional CIFAR-10 showcased DDSM's adaptability to guided diffusion. Here, we constructed a novel slimmable supernet, incorporating class labels as conditions. During inference, we leveraged classifier guidance. By applying the strategy devised for the unconditional CIFAR-10 to this supernet, we maintained the same level of FLOPs efficiency and preserved the FID score.
>
> These experiments underscore the robustness and versatility of our DDSM. They also show the good transferability of DDSM across different settings of the same dataset. To save computational cost, one could search on a small proxy dataset (for example, low-resolution data) and then apply the result to high-resolution data. Moreover, it's worth noting that the performance of DDSM on each dataset could potentially be further enhanced by restarting the search process individually for them.
>
> |  | CelebA-HQ 128x128 |  | conditional CIFAR-10 |  |
> | --- | --- | --- | --- | --- |
> |  | ADM | +DDSM | ADM | +DDSM |
> | FID | 7.53 | 7.71 | 2.48 | 2.52 |
> | GLOPs | 194.00 | 76.18 | 12.14 | 6.20 |

---

> > ### Comment · Reviewer_339s · 2023-11-21
> >
> > While there is slight degradation in FID after applying DDSM, I think it's acceptable given the improved efficiency. In particular, I like how the step strategy on lower resolution data transfer well to higher resolution data. Perhaps, it is possible to warm-start the strategy search for higher resolution data from the step strategy on lower resolution data. In any case, this addresses all my concerns, and I have raised the score to marginal accept.

---

### Official Review · Reviewer_GL7V · 2023-11-01

**Soundness:** 3 good
**Presentation:** 2 fair
**Contribution:** 3 good
**Rating:** 6
**Confidence:** 3

**Summary:**

This paper proposes Denoising Diffusion Step-aware Models (DDSM), which utilize variable-sized neural networks for different steps of the diffusion generative process. The key insight is that diffusion steps have varying importance, so uniformly allocating computational resources is inefficient. The method trains a slimmable UNet that can be flexibly pruned to different capacities. An evolutionary search then determines the optimal per-step network size to balance efficiency and performance. Experiments demonstrate substantial computational savings on CIFAR-10, CelebA-HQ, LSUN-bedroom, AFHQ, and ImageNet versus conventional diffusion models, with minimal quality loss.

**Strengths:**

- The core ideas are technically sound and offer a unique perspective on accelerating diffusion models.
- Using evolutionary search to determine step-wise network requirements.
- Compatibility with existing methods like DDIM.

**Weaknesses:**

- Using a different network at different timestep has been explored before, such as in e-diff-i.
- The compatibility claims with DDIM and latent diffusion are fairly cursory. More detailed experiments showing accelerated performance combining these methods could better showcase modularity.
- The evolutionary search itself requires non-trivial compute resources. Analysis of the search costs and scalability could be insightful.

**Questions:**

You claim compatibility with methods like DDIM and latent diffusion, but details are limited. Could you provide in-depth quantitative experiments demonstrating accelerated performance when combining DDSM with these existing diffusion acceleration techniques?

The search cost and scalability of the evolutionary algorithm is unclear. Could you analyze the computational requirements of the search procedure and discuss how it scales with factors like dataset size?

---

> ### Author Response · Authors · 2023-11-19
>
> **Discussion on EDiff-I**
>
> While eDiff-I [6] adopts multiple expert models at each step to improve image quality, it does not enhance efficiency, as all these expert models are uniformly sized. Our method distinguishes itself by utilizing models of varying sizes for acceleration purposes. Moreover, unlike eDiff-I, which manually assigns models to each step, we introduce an evolutionary search approach. This innovation not only automates model allocation but also optimizes the process, potentially yielding more effective outcomes.
>
> [6] eDiff-I: Text-to-Image Diffusion Models with an Ensemble of Expert Denoisers, Balaji et al..
>
> **Details of combining DDIM and Latent Diffusion**
>
> For the DDIM experiment, we adopted the weights from the 1000-step ADM model and applied the DDIM sampling schedule. As indicated in Table 4, our method significantly boosts DDIM's speed by 48%, 45%, 53%, and 62% for 10, 50, 100, and 1000 steps, respectively. This suggests that while additional steps enhance generation quality, they also introduce excess computational load. Our approach proves increasingly beneficial as the number of steps increases.
>
> In our latent diffusion experiment, we employed the AutoEncoderKL of SD1.4 to transform images into latent vectors, upon which our DDSM was trained. To adapt to the reduced spatial size, we modified the U-Net downsampling from 4 to 2. Our DDSM achieved a 60% acceleration in latent diffusion methods.
>
> **Analysis of the evolutionary search**
>
> A detailed discussion of search costs can be found in **Section B** of our appendix. In deep learning, models like StableDiffusion and Imagen are usually trained once and then used repeatedly. Though these models require substantial training time, they efficiently generate a large number of images upon completion. Our DDSM aims to reduce the inference costs in diffusion models by employing a step-aware pruning strategy, achieving up to 76% faster performance. Despite DDSM incurring higher initial training and search costs (about 2-3 times the training cost of standard models, with search costs similar to training a conventional model), these are one-time expenses. The specific time costs are detailed in Table B.
>
> Finally, I sincerely appreciate the valuable insights provided by the reviewers and warmly welcome any further criticisms or suggestions. Your timely feedback is crucial for the enhancement of this paper. I look forward to your valuable comments.

---

> > ### Comment · Area_Chair_cg3y · 2023-11-20
> > **Please respond to the authors' rebuttal**
> >
> > Dear reviewer,
> >
> > The window for interacting with authors on their rebuttal is closing on Wednesday (Nov 21st). Please respond to the authors' rebuttal as soon as possible, so that you can discuss any agreements or disagreements. Please acknowledge that you have read the authors' comments, and explain why their rebuttal does or does not change your opinion and score.
> >
> > Many thanks,
> > Your AC

---

### Author Response · Authors · 2023-11-19
**To all reviewers**

**Comparison with Diff-Pruning**

We appreciate the insightful inquiries from reviewers GL7V and buDR regarding the comparison of our work with the recent Diff-pruning [1]. Here, we elucidate the differences and provide an empirical comparison.
Diff-pruning employs a Taylor expansion over pruned timesteps to effectively reduce computational overhead. Reviewer buDR insightfully noted that this approach primarily focuses on pruning the entire network. Our method, in contrast, adopts a strategy of adaptively pruning the network at various timesteps. This approach allows for a more nuanced and efficient solution.
For a clearer understanding of these differences, we conducted an empirical comparison with Diff-pruning [1]. The following table showcases our experiment on CelebA 64x64 using 100 steps. Our DDSM not only achieves a FID comparable to Diff-pruning but also demonstrates superior efficiency.

|  | FID (Lower is better) | GFLOPS (Lower is better) |
| --- | --- | --- |
| Baseline | 6.48 | 49.9 |
| Diff-pruning | 6.24 | 26.6 |
| DDSM | **6.04** | **19.6** |

We would like to express our regret for not including a comparison with Diff-pruning in our initial submission to ICLR. At that time, NeurIPS had not released its accepted paper list, and our awareness of [1] was limited to its presence on arXiv. We sincerely apologize for this oversight and appreciate the chance to address it now.

[1] Structural Pruning for Diffusion Models, Fang et al., NeurIPS, 2023.

**Combining with more recent sampling schedulers**

Both Reviewer 339 and Reviewer buDR have expressed concerns regarding the compatibility of our DDSM with recent sampling schedulers, such as DPM [2], DPM++ [3], uniPC [4], and EDM [5]. These samplers are primarily focused on devising novel noise schedules to enhance the performance of diffusion models. Our method, which adaptively prunes at different steps, is theoretically compatible with these approaches. Among these samplers, we have chosen EDM for our experiments, as we believe this selection could demonstrate our method's suitability for similar methods.
In the subsequent experiment, we kept the pretrained weights of DDSM. We then replaced the original DDPM scheduler with EDM's Heun Discrete 2nd order method and initiated a new search process. The results, as illustrated in the table, show that DDSM effectively integrates with EDM on the unconditional CIFAR-10 dataset, achieving 45% and 56% total TFLOPs saving for 50 steps and 100 steps. Note that, the quality can be further improved when retraining the network with EDM’s setting.

|  | EDM50 | EDM50+DDSM | EDM100 | EDM100+DDSM |
| --- | --- | --- | --- | --- |
| FID | 3.65 | 3.61 | 3.41 | 3.49 |
| Total TFLOPs | 0.61 | 0.34 | 1.22 | 0.54 |

[2] DPM-Solver: A Fast ODE Solver for Diffusion Probabilistic Model Sampling in Around 10 Steps, Lu et al..

[3] DPM-Solver++: Fast Solver for Guided Sampling of Diffusion Probabilistic Models, Lu et al..

[4] UniPC: A Unified Predictor-Corrector Framework for Fast Sampling of Diffusion Models, Zhao et al..

[5] Elucidating the Design Space of Diffusion-Based Generative Models, Karras et al..

---

### Meta-Review · Area_Chair_cg3y · 2023-12-11

**Metareview:**

During the author-reviewer discussion phase the authors have addressed several concerns raised by the reviewers. Among other things, the authors have added results on larger resolution datasets, comparisons to recent prior work, results that demonstrate compatibility with recent sampling schedulers, and discussed how pruning and quantization approaches each have their own merit. Given the consensus of the reviewers for acceptance, the AC’s recommendation is to accept this paper.

**Justification For Why Not Higher Score:**

The reviewers clearly think this paper's proposed method is of interest to the community. However, it is a little unclear how well the evolutionary search algorithm scales to higher resolutions, although the latest results provide some evidence that a search strategy that was obtained after training on a lower resolution version of the dataset can be transferred to a model trained on a higher resolution version of the dataset. Having more clarity on the scalability aspect would have made the paper more impactful and in that case I would have considered it for a spotlight.

**Justification For Why Not Lower Score:**

All reviewers are in favor of acceptance and most concerns were addressed during the rebuttal period.

---

### Decision · Program_Chairs · 2024-01-16

Accept (poster)